# VAEM: a Deep Generative Model for Heterogeneous Mixed Type Data

Chao Ma [1]   Sebastian Tschiatschek [2]   Richard Turner [1 2]   José Miguel Hernández-Lobato [1 2]   Cheng Zhang [2]

## Abstract

Missing data imputation methods based on deep generative models often perform poorly in real-world applications, due to the heterogeneity of natural data sets. Heterogeneity arises from data containing different types of features (categorical, ordinal, continuous, etc.) and features of the same type having different marginal distributions. We propose an extension of variational autoencoders (VAEs) called VAEM to handle such heterogeneous data. We develop a corresponding efficient inference method, provide extensions and demonstrate the performance of VAEM in missing data imputation tasks. Our results show that VAEM broadens the range of real-world applications where deep generative models can be successfully deployed.

## 1. Introduction

Variational Autoencoders (VAEs) (Kingma & Welling, 2013) are powerful methods for learning low-dimensional representations in high-dimensional data, making them promising tools for enabling downstream tasks such as missing data imputation under uncertainty (Ma et al., 2018; Mattei & Frellsen, 2018; Gong et al., 2019).

However, VAEs are typically applied in standard settings in which each data dimension has similar type and similar statistical properties (e.g. consider the pixels of an image). On the contrary, many real-world datasets contain variables with different types. For instance, in healthcare applications a patient record may contain demographic information such as nationality which is of categorical type, height which is positive and continuous, and lab test results consists of images or time series.

Naively applying vanilla VAEs to such mixed type hetero-

geneous data can lead to unsatisfying results. The reason for this is that it requires the use of different likelihood functions (e.g. Gaussian likelihoods for real valued variables and Bernoulli likelihoods for binary variables). In this case, the contribution that each likelihood makes to the training objective can be very different, leading to challenging optimization problems (Kendall et al., 2018) in which some data dimensions may be poorly-modeled in favor of others. Figure 1 (c) shows an example in which a vanilla VAE fits some of the categorical variables, but performs poorly on the continuous ones.

In this paper, we present VAEM, a novel deep generative model for heterogeneous mixed type data which alleviates the limitations of VAEs discussed above (See Section 2). We carefully study the data generation and missing data imputation quality of VAEM comparing with a number of existing VAE approaches and baselines on 5 different datasets. Our results show that VAEM can model mixed type data more successfully than other baselines.

## 2. VAE for heterogeneous mixed type data

In this section, we describe our proposed method, Variational Auto-encoder for heterogeneous mixed type data (VAEM), which is a two stage model developed for such heterogeneous mixed type data. .

In order to properly handle mixed type data with heterogeneous marginals, our proposed method fits the data in a two-stage process. As shown in Figure 2(b), in the first stage we fit a different VAE independently to each data dimension $x_{nd}$. We call the resulting $D$ models *marginal VAEs*. Then, in the second stage, in order to capture the inter-variable dependencies, a new multi-dimensional VAE, called the *dependency network*, is build on top of the latent representations provided by the first-stage encoders. $D$ denotes the dimension of the observations and $N$ the number of data points with $x_{nd}$ being the $d$th dimension of the $n$th point. We present the details below.

***Stage one: training individual marginal VAEs to each single variable.*** In the first stage, we focus on modeling the marginal distributions of each variable by training $D$ individual VAEs $p_{\theta_d}(x_{nd}) = \mathbb{E}_{p(z_{nd})} p_{\theta_d}(x_{nd}|z_{nd})$, $\forall d \in \{1, 2, ..., D\}$ independently, i.e. each one is trained to fit a

---
[*]Equal contribution  [1]Department of Engineering, University of Cambridge, Cambridge, UK [2]Microsoft Research, Cambridge, UK. Correspondence to: Chao Ma <cm905@cam.ac.uk>, Cheng Zhang <Cheng.Zhang@microsoft.com>.

*Presented at the first Workshop on the Art of Learning with Missing Values (Artemiss) hosted by the $37^{th}$ International Conference on Machine Learning (ICML).* Copyright 2020 by the author(s).

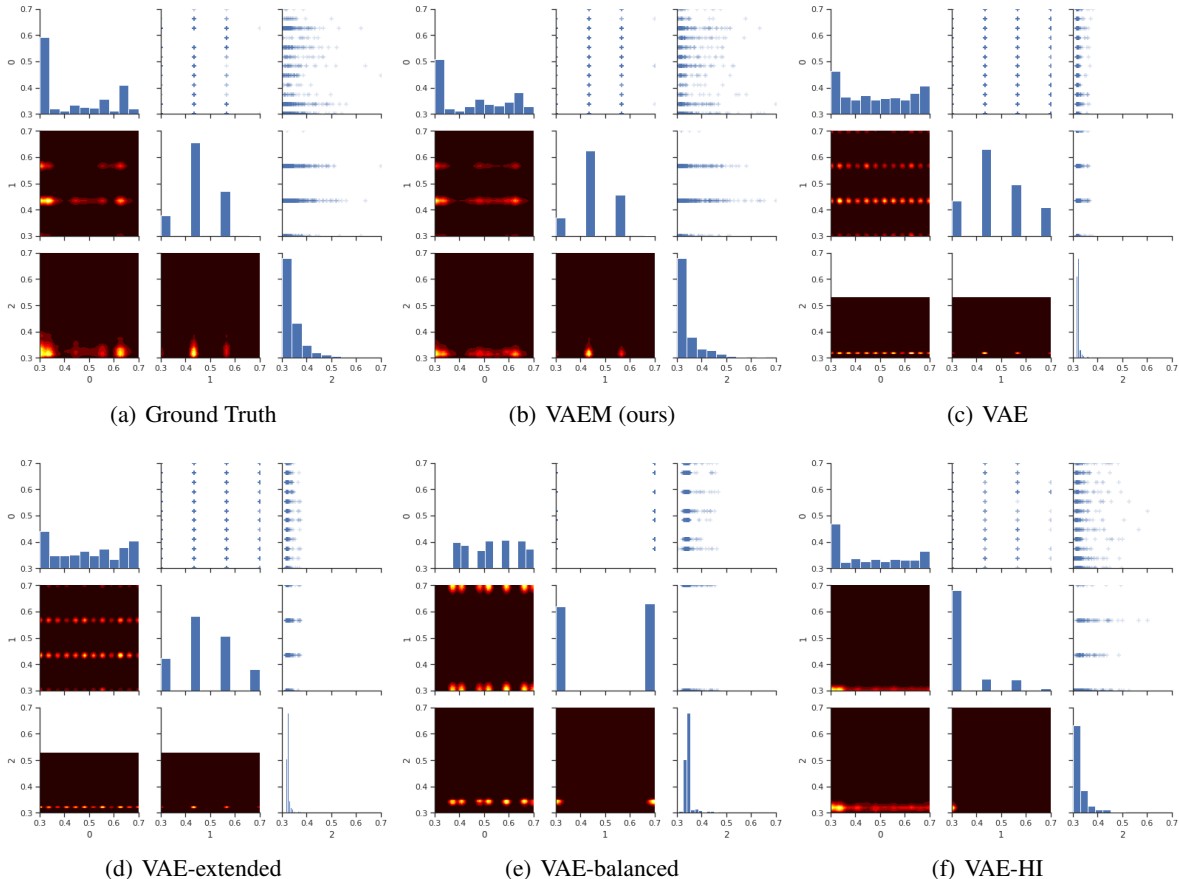

Figure 1. Pair plots of 3-dimensional data generated using five different models, fitted to the Bank dataset. Those models are defined in Section 3.1. Within each subfigure, diagonal plots show marginal histograms for each variable. Plots located above the diagonal shows sample scatter plots for each variable pair. Plots located below the diagonal show heat maps identifying regions of high-probability density for each variable pair. For visualization, categorical variables are mapped to a grid of evenly spaced points in the interval $[0, 1]$. Unlike the other baselines, VAEM can correctly capture both continuous and discrete variables correctly.

single dimension $x_{nd}$ from the dataset:

$$(\theta_d^\star, \phi_d^\star) = \arg\max_{\theta_d, \phi_d}$$

$$\sum_n \mathbb{E}_{q_{\phi_d}(z_{nd}|x_{nd})} \log \frac{p_{\theta_d}(x_{nd}, z_{nd})}{q_{\phi_d}(z_{nd}|x_{nd})} \quad \forall d \in \{1, 2, ..., D\},$$

(1)

where $p(z_{nd})$ is the standard Gaussian prior and $q_{\phi_d}(z_{nd}|x_{nd})$ is the encoder of the $d$-th marginal VAE. To specify the likelihood terms $p_{\theta_d}(x_{nd}|z_{nd})$, we use Gaussian likelihoods for continuous data and categorical likelihoods with one-hot representation for categorical data. The case of other variable types is discussed in Appendix C.1.

Note that Equation 1 contains $D$ independent objectives. Each VAE $p_d(x_{nd}; \theta_d)$ is trained independently and is only responsible for modeling the individual statistical properties of a single dimension $x_{nd}$ from the dataset. Thus, we assume that $z_{nd}$ is a scalar without loss of generality, although it would be trivial to use a multi-dimensional $\mathbf{z}_{nd}$

instead. Each marginal VAE can be trained independently until convergence (Dai & Wipf, 2019), hence avoiding the optimization issues of vanilla VAEs. We then fix the parameters of each marginal VAEs to be $\theta_d^\star$.

**Stage two: training a dependency network to connect the marginal VAEs.** In the second stage, we model the inter-variable statistical dependencies by training a new multi-dimensional VAE $p_\psi(\mathbf{z}) = \mathbb{E}_{p(\mathbf{h})} p_\psi(\mathbf{z}|\mathbf{h})$, called the *dependency network*, built on top of the latent representations $\mathbf{z}$ provided by the encoders of the marginal VAEs in the first stage. Here, $\mathbf{h}$ are the latent variables for the dependency network. Specifically, we train $p_\psi(\mathbf{z})$ as follows:

$$\mathbf{x}_{\text{data}} \sim p_{\text{data}}(\mathbf{x}), \tag{2}$$

$$z_d \sim q_{\phi_d}(z_d|x_{\text{data},d}), \quad \forall d \in \{1, ..., D\}, \tag{3}$$

$$\Delta(\psi, \lambda) \propto \nabla_{(\psi, \lambda)} \mathbb{E}_{q_\lambda(\mathbf{h}|\mathbf{z}, \mathbf{x}_{\text{data}})} \log \frac{p_\psi(\mathbf{z}, \mathbf{h})}{q_\lambda(\mathbf{h}|\mathbf{z}, \mathbf{x}_{\text{data}})}. \tag{4}$$

The above procedure effectively disentangles the heteroge-

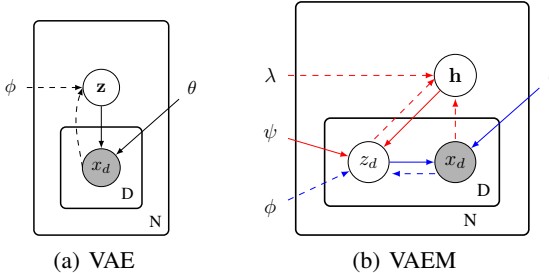

(a) VAE        (b) VAEM

*Figure 2.* Graphical representations of the vanilla VAE and our proposed VAEM. Note that in this graph, solid arrows denote decoders, and dashed arrows denote encoders. In the first stage of training a VAEM, each individual VAE (blue arrows) $p_{\theta_d}(x_{nd}) = \mathbb{E}_{p(z_{nd})} p_{\theta_d}(x_{nd}|z_{nd}), \quad \forall d \in \{1, 2, ..., D\}$ is trained independently on each variable $x_d$. Then, the dependency network (red arrows) $p_{\mathbf{z}}(\mathbf{z}_n; \psi) = \mathbb{E}_{p(\mathbf{h}_n)} p(\mathbf{z}_n|\mathbf{h}_n, \psi)$ is trained on top of the latent representations $\mathbf{z}_n$.

neous marginal properties of mixed type data (modelled by the marginal VAEs), from the inter-variable dependencies (modelled by the dependency network). We call our model *VAE for heterogeneous mixed type data (VAEM)*.

After training the marginal VAEs and dependency network, our final generative model is given by

$$p_\theta(\mathbf{x}) = \mathbb{E}_{(\mathbf{z},\mathbf{h}) \sim p(\mathbf{h}) \prod_d p_\psi(z_d|\mathbf{h})} \left[ \prod_d p_{\theta_d}(x_d|z_d) \right]. \quad (5)$$

To handle complicated statistical dependencies, we use the VampPrior (Tomczak & Welling, 2017), which specifies a mixture of Gaussians (MoGs) as the prior distribution for the high-level latent variable, i.e., $p(\mathbf{h}) = \frac{1}{K} \sum_k q_\lambda(\mathbf{h}|\mathbf{u_k})$, where $K \ll N$ and the $\{\mathbf{u_k}\}$ are a subset of data points.

### 2.1. Partial dependency network for missing data

The amortized inference network of VAEM (Section 2) cannot handle partially observed data, since the number of observed variables $\mathbf{x}_O$ might vary across different data instances. Inspired by the Partial VAE (Ma et al., 2018), we apply a PointNet to build a partial inference network in the dependency VAE that infers $\mathbf{h}$ from partial observations.

During the first stage, we estimate each marginal VAE with only the observed samples for that dimension. For the second stage, we need a dependency VAE that can handle partial observations. Similarly as in the partial-VAE (Ma et al., 2018), in the presence of missingness, the dependency VAE specifies $p_{\mathbf{z}_O}(\mathbf{z}_O; \psi) = \mathbb{E}_{p(\mathbf{h})} \prod_{d \in O} p_\psi(z_d|\mathbf{h})$. This is trained by maximizing the partial ELBO:

$$\mathbb{E}_{q_\lambda(\mathbf{h}|\mathbf{z}_O, \mathbf{x}_O)} \log \frac{\prod_{d \in O} p_\psi(z_d|\mathbf{h}) p(\mathbf{h})}{q_\lambda(\mathbf{h}|\mathbf{z}_O, \mathbf{x}_O)},$$
$$z_d \sim q_d(z_d|x_{\text{data},d}, \phi_d) \;\; \forall d \in O, \;\; \mathbf{z}_O = \{z_d|d \in O\} \quad (6)$$

where $\mathbf{h}$ is the latent variable of the dependency network, $q_\lambda(\mathbf{h}|\mathbf{z}_O, \mathbf{x}_O)$ is a set-function, the so-called *partial inference net*. Essentially, for each feature in $\mathbf{x}_O$, the input to the partial inference net is first modified as $\mathbf{s}_O := \{v \times \mathbf{e}_v | v \in \mathbf{z}_O \cup \mathbf{x}_O\}$ using element-wise multiplication, and $\mathbf{e}_v$ is a *feature embedding*[1]. $\mathbf{s}_O$ is then fed into a *feature map* (a neural network) $l(\cdot) : \mathbb{R}^M \to \mathbb{R}^K$, where $M$ and $K$ is the dimension of the feature embedding and the feature map, respectively. Finally, we apply a permutation invariant aggregation operation $g(\cdot)$, such as summation. In this way, $q_\lambda(\mathbf{h}|\mathbf{z}_O, \mathbf{x}_O)$ is invariant to the permutations of the elements of $\mathbf{x}_O$, and $\mathbf{x}_O$ can have arbitrary length.

**Approximate conditional data generation**   Once the marginal VAEs and the partial dependency network are trained, we can generate conditional samples that approximate $p_\theta(\mathbf{x}_U|\mathbf{x}_O)$ by the following inference procedure: first, the latent representations $z_d, d \in \mathcal{O}$ for the observed variables are inferred. With this representation, we use the partial inference network to infer $\mathbf{h}$, which is the latent code for the second stage VAE. Given $\mathbf{h}$, we can generate the $z_s, s \in \mathcal{U}$ which are the latent code for the unobserved dimensions and then generate the $x_s$.

## 3. Experiments

### 3.1. Baselines and datasets

In the experiments, we consider a number of baselines. All VAE baselines use the partial inference network and the discriminator specified in Section B. Moreover, all baselines are equipped with a MoG priors (Section 2). Our baselines include:

- Heterogeneous-Incomplete VAE (Nazabal et al., 2018). We match the dimensionality of latent variables to be the same as our VAEM. We denote this by `VAE-HI`.
- VAE: A vanilla VAE equipped with a VampPrior (Tomczak & Welling, 2017). The number of latent dimensions is the same as in the second stage ($\mathbf{h}$) of VAEM. We denote this by `VAE`.
- VAE with extended latent dimension: same as the `VAE`, but with the latent dimension increased to be the same as VAEM (sum of the dimensions of $\mathbf{h}$ and $\mathbf{z}$). We denote this by `VAE-extended`.
- VAE with balanced likelihoods. This baseline automatically equal the scale of each likelihood term of the different variable types, by multiplying each likelihood term with an adaptive constant (Appendix C.1). We denote this baseline by `VAE-balanced`.

We use the same pool of mixed-type datasets in all tasks:

---

[1]If $v$ is a non-continuous variable such as categorical, the operation $v \times \mathbf{e}_v$ is performed on the one-hot representation of $v$, as detailed in Appendix C.1

- Two standard UCI datasets: Boston housing and energy efficiency (Dheeru & Karra Taniskidou, 2017);
- Two relatively large real-world datasets: Bank marketing;(Moro et al., 2011) and Avocado sales prediction.
- A real-world medical dataset: MIMIC III (Johnson et al., 2016), the largest public medical dataset for intensive care.

Details including model details, hyperparameters and data processing can be found in Appendix C.

*Table 1.* Data generation quality by average test NLL per variable, with standard errors as error bars

| Method | Ours | VAE | VAE-balanced | VAE-extended | VAE-HI |
|---|---|---|---|---|---|
| Bank | **-1.15±0.09** | 2.09±0.04 | 0.72±0.01 | 2.06±0.00 | -0.72±0.00 |
| Boston | **-2.16±0.01** | -1.69±0.01 | 0.38±0.01 | -1.61±0.02 | 2.11±0.01 |
| Avocado | **-0.16±0.00** | 0.04±0.00 | 1.32±0.01 | 0.04±0.00 | 0.04±0.00 |
| Energy | -1.28±0.09 | **-1.47±0.07** | 0.69±0.02 | -1.46±0.08 | 0.16±0.00 |
| MIMIC | **-1.01±0.00** | 0.08±0.00 | 0.69±0.00 | 0.08±0.00 | 0.08±0.00 |
| Avg. Rank | **1.40±0.36** | 2.60±0.61 | 4.40±0.36 | 3.00±0.40 | 3.00±0.57 |

### 3.2. Mixed type data generation

In this task, we evaluate the quality of our generative model in terms of mixed type data generation. For all datasets, we first train the models and then quantitatively compare their performance using a 90%-10% train-test split. All experiments are repeated 5 times.

**Visualization by pair plots** In deep generative models, the data generation quality is indicative of how well the model describes the data. Thus, we first visualize the data generated by each model on a representative dataset: Bank marketing. By comparing the plots in the diagonals of Figure 1 (a) and Figure 1 (c), we notice that vanilla VAE is able to describe the marginal distribution of the second categorical variable. However, it fails to mimic the behaviour of the third variable. Note that this variable (Figure 1 (a)), which corresponds to the "duration" feature of the dataset, has a heavy tail behaviour, which is ignored by vanilla VAE. On the other hand, although the VAE-balanced model and VAE-HI (Figure 1 (e) (f)) can partially describe this heavy-tail behaviour, it fails to model the marginal distribution of second categorical variable well. Unlike the baselines, our VAEM model (Figure 1 (b)) is able to accurately describe the marginals and joint distributions for both categorical and heavy-tailed continuous distribution.

**Quantitative evaluation on all datasets** To evaluate the data generation quality quantitatively, we compute the marginal negative log-likelihood (NLL) of the models on the test sets. Note that all NLL numbers are divided by the number of variables of the dataset. As shown in Table 1, VAEM can consistently generate realistic samples, and on

*Table 2.* Conditional data generation quality under random missing entries. Test NLL per variable, with standard errors as error bars.

| Method | Ours | VAE | VAE-balanced | VAE-extended | VAE-HI |
|---|---|---|---|---|---|
| Bank | **-1.21±0.12** | 2.09±0.00 | 0.68±0.00 | 2.09±0.00 | -0.83±0.01 |
| Boston | **-2.18±0.03** | -1.66±0.02 | 0.37±0.00 | -1.67±0.01 | 1.58±0.01 |
| Avocado | **-0.15±0.00** | 0.04±0.00 | 1.33±0.00 | 0.04±0.00 | 0.04±0.00 |
| Energy | -1.30±0.05 | **-1.50±0.06** | 0.67±0.01 | -1.50±0.06 | 0.13±0.00 |
| MIMIC | **-1.10±0.00** | 0.08±0.00 | 0.57±0.00 | 0.08±0.00 | 0.08±0.00 |
| Avg. Rank | **1.40±0.36** | 2.60±0.61 | 4.40±0.38 | 2.30±0.44 | 3.00±0.57 |

average significantly outperforms other baselines.

### 3.3. Mixed type probabilistic missing data imputation

An important aspect of generative models is their ability to perform probabilistic missing data imputation (conditional data generation) (Ma et al., 2018; Mattei & Frellsen, 2018; Gong et al., 2019). That is, given a data instance, to infer the posterior distribution of unobserved variables $\mathbf{x}_U$ given observed $\mathbf{x}_O$. For all baselines evaluated in this task, we train the partial version of them (i.e., generative + partial inference net (Ma et al., 2018)). To train the partial models, we randomly sample 90% of the dataset to be the training set, and assume that a random fraction (uniformly sampled between 0% and 99%) of feature values are missing each epoch during training. Then, during test time, we assume that 50% of the test set is observed, and use generative models to make inference regarding the rest of unobserved data. Since all inference are probabilistic, we report the negative test NLLs on unobserved data, as opposed to imputation RMSE typically used in the literature.

Results are summarized in Table 2, where all NLL values have been divided by the number of observed variables. We repeat our experiments for 5 runs and report standard errors. Note that the automatic balancing strategy **VAE-balanced** almost always deteriorates the performance. By contrast, Table 2 shows that our proposed method is very robust, yielding significantly better performance than baselines.

## 4. Conclusion

We proposed VAEM, a novel two stage deep generative modelthat can handle mixed type data with heterogeneous marginals and missing data. VAEM sidesteps the problems arising from heterogeneous data. Efficient amortized inference methods and extensions are proposed. Experiments yield promising results, indicating that VAEM is useful for real-world applications of deep generative models.

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

# A. Additional Derivations

## A.1. Information reward approximation for hierarchical generative models in the present of missing latent variable

We consider the estimation of the following *information reward* function

$$R_I(\mathbf{x}_i, \mathbf{x}_O) = \mathbb{E}_{\mathbf{x}_i \sim p(\mathbf{x}_i|\mathbf{x}_O)} \mathbb{KL}\left[p(\mathbf{x}_\Phi|\mathbf{x}_i, \mathbf{x}_O) \,\|\, p(\mathbf{x}_\Phi|\mathbf{x}_O)\right]$$

Using our proposed VAEM method (the partial VAEM version in 2.1). The VAEM is a hierarchical generative model trained by the two-stage procedure described in the paper. Conditional inference of VAEM of missing data follows the following sampling process:

$$
\begin{aligned}
z_d &\sim q_d(z_d|x_{,d}, \mathbf{\Phi}_d) \ \ \forall d \in O, \ \ \mathbf{z}_O = \{z_d | d \in O\} \\
\mathbf{h} &\sim q_\lambda(\mathbf{h}|\mathbf{z}_O) \\
z_s &\sim p_\psi(z_s|\mathbf{h}) \ \ \forall s \in U, \ \ \mathbf{z}_U = \{z_s | s \in U\} \\
x_s &\sim p_\theta(x_s|\mathbf{z}_U, \mathbf{z}_O) \ \ \forall s \in U, \ \ \mathbf{x}_U = \{x_s | s \in U\}
\end{aligned}
$$

Note that for compactness, we omitted the notation for input $\mathbf{x}_O$ and $\mathbf{x}_i$ to the all partial inference nets $q_\lambda$. Where $\mathbf{z}_O$ is the observed latent variables of marginal VAEs, and $\mathbf{z}_U$ are unobserved. We will use this VAEM to estimate any probabilistic quantities in information reward A.1.

Applying the chain rule of KL-divergence on the term $\mathbb{KL}\left[p(\mathbf{x}_\Phi|\mathbf{x}_i, \mathbf{x}_O) \,\|\, p(\mathbf{x}_\Phi|\mathbf{x}_O)\right]$, we have:

$$
\begin{aligned}
&\mathbb{KL}(p(\mathbf{x}_\Phi|\mathbf{x}_i, \mathbf{x}_O)||p(\mathbf{x}_\Phi|\mathbf{x}_O)) \\
&= \mathbb{KL}(p(\mathbf{x}_\Phi, \mathbf{z}_i, \mathbf{z}_O, \mathbf{h}|\mathbf{x}_i, \mathbf{x}_O)||p(\mathbf{x}_\Phi, \mathbf{z}_i, \mathbf{z}_O, \mathbf{h}|\mathbf{x}_O)) \\
&\quad - \mathbb{E}_{\mathbf{x}_\Phi \sim p(\mathbf{x}_\Phi|\mathbf{x}_i, \mathbf{x}_O)}\left[\mathbb{KL}(p(\mathbf{z}_\Phi, \mathbf{z}_i, \mathbf{z}_O, \mathbf{h}|\mathbf{x}_\Phi, \mathbf{x}_i, \mathbf{x}_O)||p(\mathbf{z}_\Phi, \mathbf{z}_i, \mathbf{z}_O, \mathbf{h}|\mathbf{x}_\Phi, \mathbf{x}_O))\right],
\end{aligned}
$$

Based on the independencies of marginal VAEs, we have $p(\mathbf{x}_\Phi, \mathbf{z}_i, \mathbf{z}_O, \mathbf{h}|\mathbf{x}_O)) = p(\mathbf{x}_\Phi, \mathbf{z}_O, \mathbf{h}|\mathbf{x}_O))p(\mathbf{z}_i)$, $p(\mathbf{z}_\Phi, \mathbf{z}_i, \mathbf{z}_O, \mathbf{h}|\mathbf{x}_\Phi, \mathbf{x}_O)) = p(\mathbf{z}_\Phi, \mathbf{z}_i, \mathbf{z}_O, \mathbf{h}|\mathbf{x}_\Phi, \mathbf{x}_O))p(\mathbf{z}_i)$.

Using again the KL-divergence chain rule on $\mathbb{KL}(p(\mathbf{x}_\Phi, \mathbf{z}_i, \mathbf{z}_O, \mathbf{h}|\mathbf{x}_i, \mathbf{x}_O)||p(\mathbf{x}_\Phi, \mathbf{z}_i, \mathbf{z}_O, \mathbf{h}|\mathbf{x}_O))$, we have:

$$
\begin{aligned}
&\mathbb{KL}(p(\mathbf{x}_\Phi, \mathbf{z}_i, \mathbf{z}_O, \mathbf{h}|\mathbf{x}_i, \mathbf{x}_O)||p(\mathbf{x}_\Phi, \mathbf{z}_i, \mathbf{z}_O, \mathbf{h}|\mathbf{x}_O)) \\
&= \mathbb{KL}(p(\mathbf{z}_i, \mathbf{z}_O, \mathbf{h}|\mathbf{x}_i, \mathbf{x}_O)||p(\mathbf{z}_i, \mathbf{z}_O, \mathbf{h}|\mathbf{x}_O)) + \mathbb{E}_{p(\mathbf{z}_\Phi, \mathbf{z}_i, \mathbf{z}_O, \mathbf{h}|\mathbf{x}_i, \mathbf{x}_O)}\mathbb{KL}(p(\mathbf{x}_\Phi|\mathbf{z}_i, \mathbf{z}_O, \mathbf{h}, \mathbf{x}_i, \mathbf{x}_O)||p(\mathbf{x}_\Phi|\mathbf{z}_i, \mathbf{z}_O, \mathbf{h}, \mathbf{x}_O)) \\
&= \mathbb{KL}(p(\mathbf{z}_i, \mathbf{z}_O, \mathbf{h}|\mathbf{x}_i, \mathbf{x}_O)||p(\mathbf{z}_i, \mathbf{z}_O, \mathbf{h}|\mathbf{x}_O)) + \mathbb{E}_{p(\mathbf{z}_\Phi, \mathbf{z}_i, \mathbf{z}_O, \mathbf{h}|\mathbf{x}_i, \mathbf{x}_O)}\mathbb{KL}(p(\mathbf{x}_\Phi|\mathbf{z}_i, \mathbf{z}_O, \mathbf{h})||p(\mathbf{x}_\Phi|\mathbf{z}_i, \mathbf{z}_O, \mathbf{h})) \\
&= \mathbb{KL}(p(\mathbf{z}_i, \mathbf{z}_O, \mathbf{h}|\mathbf{x}_i, \mathbf{x}_O)||p(\mathbf{z}_i, \mathbf{z}_O, \mathbf{h}|\mathbf{x}_O)).
\end{aligned}
$$

Note that the last two equalities does not hold for the discriminative version of VAEM described in Section B. Fortunately, $\mathbb{E}_{\mathbf{x}_i \sim p(\mathbf{x}_i|\mathbf{x}_O)}\mathbb{KL}(p(\mathbf{x}_\Phi|\mathbf{z}_i, \mathbf{z}_O, \mathbf{h}, \mathbf{x}_i, \mathbf{x}_O)||p(\mathbf{x}_\Phi|\mathbf{z}_i, \mathbf{z}_O, \mathbf{h}, \mathbf{x}_O)) = 0$ still holds for the discriminative version, hence we will still arrive at the same result.

The KL-divergence term in the reward formula is now rewritten as follows,

$$
\begin{aligned}
&\mathbb{KL}(p(\mathbf{x}_\Phi|\mathbf{x}_i, \mathbf{x}_O)||p(\mathbf{x}_\Phi|\mathbf{x}_O)) \\
&= \textcolor{blue}{\mathbb{KL}(p(\mathbf{z}_i, \mathbf{z}_O, \mathbf{h}|\mathbf{x}_i, \mathbf{x}_O)||p(\mathbf{z}_i, \mathbf{z}_O, \mathbf{h}|\mathbf{x}_O))} \\
&\quad - \mathbb{E}_{\mathbf{x}_\Phi \sim p(\mathbf{x}_\Phi|\mathbf{x}_i, \mathbf{x}_O)}\left[\mathbb{KL}(p(\mathbf{z}_\Phi, \mathbf{z}_i, \mathbf{z}_O, \mathbf{h}|\mathbf{x}_\Phi, \mathbf{x}_i, \mathbf{x}_O)||p(\mathbf{z}_\Phi, \mathbf{z}_i, \mathbf{z}_O, \mathbf{h}|\mathbf{x}_\Phi, \mathbf{x}_O))\right].
\end{aligned}
$$

For the term in blue, we have:

$$
\begin{aligned}
&\textcolor{blue}{\mathbb{KL}(p(\mathbf{z}_i, \mathbf{z}_O, \mathbf{h}|\mathbf{x}_i, \mathbf{x}_O)||p(\mathbf{z}_i, \mathbf{z}_O, \mathbf{h}|\mathbf{x}_O))} \\
&= \mathbb{KL}(p(\mathbf{z}_i, \mathbf{z}_O|\mathbf{x}_i, \mathbf{x}_O)||p(\mathbf{z}_O|\mathbf{x}_O)p(\mathbf{z}_i)) \\
&\quad + \mathbb{E}_{\mathbf{z}_i, \mathbf{z}_O \sim p(\mathbf{z}_i, \mathbf{z}_O|\mathbf{x}_i, \mathbf{x}_O)}\left[\mathbb{KL}\left(p(\mathbf{h}|\mathbf{z}_i, \mathbf{z}_O)||p(\mathbf{h}|\mathbf{z}_O)\frac{p(\mathbf{z}_i)}{p(\mathbf{z}_i|\mathbf{x}_O)}\right)\right] \\
&= \mathbb{KL}(p(\mathbf{z}_i|\mathbf{x}_i)||p(\mathbf{z}_i)) + \mathbb{E}_{\mathbf{z}_i, \mathbf{z}_O \sim p(\mathbf{z}_i, \mathbf{z}_O|\mathbf{x}_i, \mathbf{x}_O)}\left[\mathbb{KL}(p(\mathbf{h}|\mathbf{z}_i, \mathbf{z}_O)||p(\mathbf{h}|\mathbf{z}_O))\right]
\end{aligned}
$$

Similarly for the term in red, we have:

$$\mathbb{KL}(p(\mathbf{z}_\Phi, \mathbf{z}_i, \mathbf{z}_O, \mathbf{h}|\mathbf{x}_\Phi, \mathbf{x}_i, \mathbf{x}_O)||p(\mathbf{z}_\Phi, \mathbf{z}_i, \mathbf{z}_O, \mathbf{h}|\mathbf{x}_\Phi, \mathbf{x}_O))$$

$$= \mathbb{KL}(p(\mathbf{z}_\Phi, \mathbf{z}_i, \mathbf{z}_O|\mathbf{x}_\Phi, \mathbf{x}_i, \mathbf{x}_O)||p(\mathbf{z}_\Phi, \mathbf{z}_O|\mathbf{x}_\Phi, \mathbf{x}_O)p(\mathbf{z}_i))$$

$$+ \mathbb{E}_{\mathbf{z}_\Phi, \mathbf{z}_i, \mathbf{z}_O \sim p(\mathbf{z}_\Phi, \mathbf{z}_i, \mathbf{z}_O|\mathbf{x}_\Phi, \mathbf{x}_i, \mathbf{x}_O)} \left[ \mathbb{KL}\left( p(\mathbf{h}|\mathbf{z}_\Phi, \mathbf{z}_i, \mathbf{z}_O)||p(\mathbf{h}|\mathbf{z}_\Phi, \mathbf{z}_O)\frac{p(\mathbf{z}_i)}{p(\mathbf{z}_i|\mathbf{x}_\Phi, \mathbf{x}_O)} \right) \right]$$

$$= \mathbb{KL}(\mathbf{z}_i|\mathbf{x}_i)||p(\mathbf{z}_i)) + \mathbb{E}_{\mathbf{z}_\Phi, \mathbf{z}_i, \mathbf{z}_O \sim p(\mathbf{z}_\Phi, \mathbf{z}_i, \mathbf{z}_O|\mathbf{x}_\Phi, \mathbf{x}_i, \mathbf{x}_O)} [\mathbb{KL}(p(\mathbf{h}|\mathbf{z}_\Phi, \mathbf{z}_i, \mathbf{z}_O)||p(\mathbf{h}|\mathbf{z}_\Phi, \mathbf{z}_O))]$$

Finally, we have:

$$R_I(\mathbf{x}_i, \mathbf{x}_O)$$

$$= \mathbb{E}_{\mathbf{x}_i \sim p(\mathbf{x}_i|\mathbf{x}_O)}\mathbb{KL}\left[ p(\mathbf{x}_\Phi|\mathbf{x}_i, \mathbf{x}_O) \,\|\, p(\mathbf{x}_\Phi|\mathbf{x}_O) \right]$$

$$= \mathbb{E}_{\mathbf{x}_i \sim p(\mathbf{x}_i|\mathbf{x}_O)}\mathbb{KL}(p(\mathbf{z}_i, \mathbf{z}_O, \mathbf{h}|\mathbf{x}_i, \mathbf{x}_O)||p(\mathbf{z}_i, \mathbf{z}_O, \mathbf{h}|\mathbf{x}_O))$$

$$- \mathbb{E}_{\mathbf{x}_i \sim p(\mathbf{x}_i|\mathbf{x}_O)}\mathbb{E}_{\mathbf{x}_\Phi \sim p(\mathbf{x}_\Phi|\mathbf{x}_i, \mathbf{x}_O)} [\mathbb{KL}(p(\mathbf{z}_\Phi, \mathbf{z}_i, \mathbf{z}_O, \mathbf{h}|\mathbf{x}_\Phi, \mathbf{x}_i, \mathbf{x}_O)||p(\mathbf{z}_\Phi, \mathbf{z}_i, \mathbf{z}_O, \mathbf{h}|\mathbf{x}_\Phi, \mathbf{x}_O))]$$

$$= \mathbb{E}_{\mathbf{x}_i, \mathbf{z}_i, \mathbf{z}_O \sim p(\mathbf{x}_i, \mathbf{z}_i, \mathbf{z}_O|\mathbf{x}_O)} \{\mathbb{KL}\left[ p(\mathbf{h}|\mathbf{z}_i, \mathbf{z}_O)||p(\mathbf{h}|\mathbf{z}_O) \right]$$

$$- \mathbb{E}_{\mathbf{x}_\Phi, \mathbf{z}_\Phi \sim p(\mathbf{x}_\Phi, \mathbf{z}_\Phi, |\mathbf{x}_O)}\mathbb{KL}\left[ p(\mathbf{h}|\mathbf{z}_\Phi, \mathbf{z}_i, \mathbf{z}_O)||p(\mathbf{h}|\mathbf{z}_\Phi, \mathbf{z}_O) \right] \}.$$

We can then plug in the VAEM model distirbutions:

$$p(\mathbf{x}_i, \mathbf{z}_i, \mathbf{z}_O|\mathbf{x}_O) = p_{\theta, \phi}(\mathbf{x}_i, \mathbf{z}_i, \mathbf{z}_O|\mathbf{x}_O)$$

$$p(\mathbf{x}_\Phi, \mathbf{z}_\Phi, |\mathbf{x}_O) = p_{\theta, \phi}(\mathbf{x}_\Phi, \mathbf{z}_\Phi, |\mathbf{x}_O)$$

$$p(\mathbf{h}|\mathbf{z}_i, \mathbf{z}_O) \approx q_\lambda(\mathbf{h}|\mathbf{z}_i, \mathbf{z}_O)$$

$$p(\mathbf{h}|\mathbf{z}_O) \approx q_\lambda(\mathbf{h}|\mathbf{z}_O)$$

$$p(\mathbf{h}|\mathbf{z}_\Phi, \mathbf{z}_i, \mathbf{z}_O) \approx q_\lambda(\mathbf{h}|\mathbf{z}_\Phi, \mathbf{z}_i, \mathbf{z}_O)$$

$$p(\mathbf{h}|\mathbf{z}_\Phi, \mathbf{z}_O) \approx q_\lambda(\mathbf{h}|\mathbf{z}_\Phi, \mathbf{z}_O)$$

Finally, the information reward is now approximated as:

$$R_I(\mathbf{x}_i, \mathbf{x}_O)$$

$$\approx \mathbb{E}_{\mathbf{x}_i, \mathbf{z}_i, \mathbf{z}_O \sim p_{\theta, \phi}(\mathbf{x}_i, \mathbf{z}_i, \mathbf{z}_O|\mathbf{x}_O)} \{\mathbb{KL}\left[ q_\lambda(\mathbf{h}|\mathbf{z}_i, \mathbf{z}_O)||q_\lambda(\mathbf{h}|\mathbf{z}_O) \right]$$

$$- \mathbb{E}_{\mathbf{x}_\Phi, \mathbf{z}_\Phi \sim p_{\theta, \phi}(\mathbf{x}_\Phi, \mathbf{z}_\Phi, |\mathbf{x}_O)}\mathbb{KL}\left[ q_\lambda(\mathbf{h}|\mathbf{z}_\Phi, \mathbf{z}_i, \mathbf{z}_O)||q_\lambda(\mathbf{h}|\mathbf{z}_\Phi, \mathbf{z}_O) \right] \}.$$

## B. Enhancing predictive performance of VAEM: training procedure

In order to enhance the predictive performance of VAEM, the following alternative factorization is proposed:

$$p_\theta(\mathbf{x}_O, \mathbf{x}_\Phi) = \mathbb{E}_{\mathbf{x}_{U \setminus \Phi}, \mathbf{h} \sim p_\theta(\mathbf{x}_{U \setminus \Phi}, \mathbf{h}|\mathbf{x}_O)}p_\gamma(\mathbf{x}_\Phi|\mathbf{x}_O, \mathbf{x}_{U \setminus \Phi}, \mathbf{h})p_\theta(\mathbf{x}_O)$$

For compactness, the notation for input $\mathbf{x}_O$ and $\mathbf{x}_i$ to the all partial inference nets $q_\lambda$ will be omitted. Note that, to train this model, we also need data samples of $\mathbf{x}_\Phi$ during training (however $\mathbf{x}_\Phi$ will not be observed during active learning task). This model is trained using the following procedure:

- Train a partial VAEM on $\mathbf{x}_O$ ($\mathbf{x}_\Phi \cap \mathbf{x}_O = \varnothing$) using the two-stage method described in Section 2. Now we have a graphical model induced by the model $p_\theta(\mathbf{x}_O)$.

- Expand the graph by adding the node $\mathbf{x}_\Phi$ to the graph. Now the joint distribution is defined as $p_\theta(\mathbf{x}_O, \mathbf{x}_\Phi) = \mathbb{E}_{\mathbf{x}_{U \setminus \Phi}, \mathbf{h} \sim p_\theta(\mathbf{x}_{U \setminus \Phi}, \mathbf{h} | \mathbf{x}_O)} p_\gamma(\mathbf{x}_\Phi | \mathbf{x}_O, \mathbf{x}_{U \setminus \Phi}, \mathbf{h}) p_\theta(\mathbf{x}_O)$. Note that no new parameters need to be introduced for the partial inference net of the dependency network $q_\lambda(\mathbf{h} | \mathbf{z}_O, \mathbf{z}_\Phi)$, since the partial inference net automatically handles inputs with different dimensionalities.

- Define the marginal VAE encoder for $x_\Phi$ as $q_d(z_\Phi | x_{n,\Phi}, \phi_\Phi) = \delta(z_\Phi - x_\Phi)$, and the decoder to be $p_d(x_{n,\Phi} | z_d, \theta_\Phi) = \delta(x_\Phi - z_\Phi)$ (i.e., both are identity deterministic mappings).

- The partial inference net parameters of the dependency network can be updated by the following procedure:

$$z_d \sim q_d(z_d | x_{\text{data}, d}, \phi_d) \ \forall d \in O \cup \Phi, \ \mathbf{z}_{O \cup \Phi} = \{z_d | d \in O \cup \Phi\}$$

$$\Delta\lambda \propto \nabla_\lambda \mathbb{E}_{q_\lambda(\mathbf{h} | \mathbf{z}_{O \cup \Phi})} \left[ \log \frac{\prod_{d \in O} p_\psi(z_d | \mathbf{h}) p(\mathbf{h})}{q_\lambda(\mathbf{h} | \mathbf{z}_{O \cup \Phi})} + \mathbb{E}_{\mathbf{x}_{U \setminus \Phi} \sim p_{\theta, \psi}(\mathbf{x}_{U \setminus \Phi} | \mathbf{h})} \log p_\gamma(\mathbf{x}_\Phi | \mathbf{x}_O, \mathbf{x}_{U \setminus \Phi}, \mathbf{h}) \right]$$

- The the parameters for $p_\gamma(\mathbf{x}_\Phi | \mathbf{x}_O, \mathbf{x}_{U \setminus \Phi}, \mathbf{h})$ can be updated by the following procedure:

$$z_d \sim q_d(z_d | x_{,d}, \phi_d) \ \forall d \in O, \ \mathbf{z}_O = \{z_d | d \in O\}$$
$$\mathbf{h} \sim q_\lambda(\mathbf{h} | \mathbf{z}_O)$$
$$z_s \sim p_\psi(z_s | \mathbf{h}) \ \forall s \in U \setminus \Phi, \ \mathbf{z}_{U \setminus \Phi} = \{z_s | s \in U \setminus \Phi\}$$
$$x_s \sim p_\theta(x_s | \mathbf{z}_U, \mathbf{z}_O) \ \forall s \in U \setminus \Phi, \ \mathbf{x}_{U \setminus \Phi} = \{x_s | s \in U \setminus \Phi\}$$
$$\gamma^\star = \arg\max_\gamma \log p_\gamma(\mathbf{x}_\Phi | \mathbf{x}_O, \mathbf{x}_{U \setminus \Phi}, \mathbf{h})$$

## C. Additional Experiment Settings

subsectionDatasets details We use the same collection of mixed type datasets in all tasks:

- Two standard UCI benchmark datasets: Boston housing (13 continuous, 1 categorical) and energy efficiency (6 continuous, 3 categorical) (Dheeru & Karra Taniskidou, 2017);
- Two relatively large real-world dataset: Bank marketing (45211 instances, 11 continuous, 8 categorical, 2 discrete); (Moro et al., 2011) and Avocado sales prediction (18249 instances, 9 continuous, 5 categorical).
- One real-world medical dataset: Medical Information Mart for Intensive Care (MIMIC III) (Johnson et al., 2016), the largest public medical dataset containing records of 21139 patients (after processing following (Harutyunyan et al., 2017)). We focus on the mortality prediction task based on 17 medical instruments (13 continuous, 4 categorical). Since the dataset is imbalanced (over 80 % of the data has mortality $= 0$), we balance the dataset by down-sampling the majority class. The time-series observations are averaged to obtain iid data points.

### C.1. Additional model specification

#### C.1.1. BASELINES: GENERAL INFORMATION

We have used the following baselines in our experiments:

- Heterogeneous-Incomplete VAE (HI-VAE) (Nazabal et al., 2018). We adopt the multi-head structure of HI-VAE and match the dimensionality of latent variables to be the same as our VAEM. HI-VAE is an important baseline, since it is motivated in a similar way as our VAEM, but all marginal VAEs are trained jointly rather as opposed to our two-stage method. We denote this by `VAE-HI`
- VAE: A vanilla VAE equipped with a VampPrior (Tomczak & Welling, 2017). The number of latent dimensions is the same as in the second stage of VAEM. We denote this by `VAE`.
- VAE with extended latent dimension: Note that the total number of latent variables of VAEM is $D + L$, where $D$ and $L$ are the dimensionalities of the data and the latent space, respectively. This baseline is like the previous one, but with the latent dimension given by $D + L$. We denote this baseline by `VAE-extended`.
- VAE with automatically balanced likelihoods. This baseline tries to automatically equal the scale of each likelihood term of the different variable types in the ELBO by multiplying each likelihood term with an adaptive constant (Appendix C.1). We denote this baseline by `VAE-balanced`.

## C.1.2. BASELINE: VAE WITH BALANCED LIKELIHOODS

This baseline is a naive strategy that tries to automatically balance the scale of the log-likelihood values of different variable types in the ELBO, by adaptively multiplying a constant before likelihood terms. More specifically, consider the variational lower bound (ELBO) of vanilla VAE:

$$\log p_\theta(\mathbf{x}) \geq \mathbb{E}_{q_\phi(\mathbf{z}|\mathbf{x})} \log \frac{p_\theta(\mathbf{x}, \mathbf{z})}{q_\phi(\mathbf{z}|\mathbf{x})}$$

$$= \sum_{s \in \mathcal{P}} \mathbb{E}_{q_\phi(\mathbf{z}|\mathbf{x})} \log \frac{p_\theta(\mathbf{x}_{s \in \mathcal{P}}, \mathbf{z})}{q_\phi(\mathbf{z}|\mathbf{x})}$$

Where $\mathcal{P}$ is the set of variable types (e.g., continuous, categorical), and $\mathbf{x}_s$ is the set of variables that belong to $s$-th type. In VAE with balanced likelihoods, we weight each likelihood terms by $\{\beta_1, \beta_2, ..., \beta_{|\mathbf{P}|}\}$:

$$\sum_{s \in \mathcal{P}} \beta_s \mathbb{E}_{q_\phi(\mathbf{z}|\mathbf{x})} \log \frac{p_\theta(\mathbf{x}_{s \in \mathcal{P}}, \mathbf{z})}{q_\phi(\mathbf{z}|\mathbf{x})}$$

Where $\sum_s \beta_s = 1$, such that:

$$\beta_s \mathbb{E}_{q_\phi(\mathbf{z}|\mathbf{x})} \log p_\theta(\mathbf{x}_s|\mathbf{z}) = \beta_t \mathbb{E}_{q_\phi(\mathbf{z}|\mathbf{x})} \log p_\theta(\mathbf{x}_t|\mathbf{z}), \;\; \forall s, t \in \mathcal{P}$$

In practice, at each epoch of training, a mini-batch $\{\mathbf{x}_j\}_{1 \leq j \leq J}$ is selected, and $\beta_s$ are estimated such that:

$$\beta_s \sum_j \mathbb{E}_{q_\phi(\mathbf{z}_j|\mathbf{x}_j)} \log p_\theta(\mathbf{x}_{j,s}|\mathbf{z}_j) = \beta_t \sum_j \mathbb{E}_{q_\phi(\mathbf{z}_j|\mathbf{x}_j)} \log p_\theta(\mathbf{x}_{j,t}|\mathbf{z}_j), \;\; \forall s, t \in \mathcal{P}$$

## C.1.3. LIKELIHOOD FUNCTION SPECIFICATION

In this paper, we consider three variable types: continuous, categorical, and discrete. For continuous and categorical variables, we follow the specification of (Nazabal et al., 2018). In other words, to specify the likelihood function of all VAE decoders $p_{\theta_d}(x_{nd}|z_{nd})$ in our paper, we use Gaussian likelihood with constant observational noises $p_{\theta_d}(x_{nd}|z_{nd}) = \mathcal{N}(x_{nd}; \mu(z_{nd}), \sigma^2)$ for continuous data; and for categorical data, we use categorical likelihood with one-hot representation $p_{\theta_d}(x_{nd}|z_{nd}) = \langle \mathbf{l}(z_{nd}), \texttt{one-hot}(x_{nd}) \rangle$, where $\mathbf{l}(z_{nd})$ is soft-max output of the decoder.

For discrete variables, we consider two different scenarios: continuous-discrete and ordinal-discrete. Continuous-discrete means that the variable is continuous by its nature, but only discretized values are recorded. For example, the salary (dollars) is a continuous variable, but in practice only discretized values (5000 dollars, 6000 dollars, etc.) are recorded. For this type of variables, we still use Gaussian likelihood, but the decoder output will be rounded to the closest discrete value. On the other hand, ordinal-discrete variables (such as ratings) are the ones with natural orderings, and the distance between each value is not known. For ordinal variables, we use ordinal regression likelihood used in (Paquet et al., 2012).

Note that the above settings are used for all models including VAEM and other baselines.

## C.1.4. PARTIAL INFERENCE NET WITH NON-CONTINUOUS INPUT

. In section 2.1, the partial inference net $q_\lambda(\mathbf{h}|\mathbf{z}_O, \mathbf{x}_O)$ is constructed based on the element-wise multiplication operation $\mathbf{s}_O := \{v \times \mathbf{e}_v | v \in \mathbf{z}_O \cup \mathbf{x}_O\}$. How is $v \times \mathbf{e}_v$ defined if $v$ is non-continuous? For categorical and ordinal-discrete variable for example, the operation $v \times \mathbf{e}_v$ is defined as

$$v \times \mathbf{e}_v := vec(\texttt{one-hot}(v) \otimes \mathbf{e}_v)$$

Where $\otimes$ is outer-product between vectors, $\texttt{one-hot}$ is the one-hot representation of the categorical/ordinal variables, and $vec(\cdot)$ is the vectorization operation of a matrix.

## C.2. Network structure and hyper parameter settings

**Network structures** All models (except for the marginal VAEs of VAEM and the decoder of HI-VAE) share the same network structures with 20 dimensional diagonal Gaussian latent variables: the generator (decoder) is a 20-50-100 fully

connected neural network with ReLU activation functions on hidden units (where $D$ is the data dimension). Note that we use sigmoid activation function for output layer, to reflect our data preprocessing (all data are normalized to between 0 and 1). One exception is the output layer of dependency network of VAEM, where we did not add any activation functions since the scales of the latent variables $z_d$ from marginal VAEs are unknown. The encoders share the same structure of $D$-500-200-40 that maps the observed data into distributional parameters of the latent space. Additionally, we use a $K = 100$ dimensional feature mapping parameterized by a single layer neural network, and $M = 10$ dimensional feature embedding for each variable. We choose the permutation invariant operator $g$ to be the summation operator. The discriminator described in section B is a neural network with two layers, each of which has 100 hidden units.

For marginal VAEs of our VAEM, we use 1-dimensional latent variable for each variable.The decoder of marginal VAEs is a 1-50-V single layer neural network, and the encoder network structure is V-50-2, where $V$ is the dimension of the corresponding variable, which is defined to be 1 if the variable is continuous. Otherwise, $V$ is the dimension of the one-hot representation. The same structure is used for the multi-head decoder structure for HI-VAE baseline.

**Hyperparameters**   To train our models, we apply Adam optimization (Kingma & Ba, 2015) with learning rate of 0.001 and a batch size of 100. When the training set is fully observed, We manually generate partially observed version of it by adding artificially missingness at random in the training dataset during training. This will help the model to learn to generate conditional data given observations. We first draw a missing rate parameter from a uniform distribution $\mathcal{U}(0, 1)$ and randomly choose variables as unobserved. This step is repeated at each iteration. We train our models for 3000 full epochs, except for Bank dataset where we used 5000 epochs. For continuous variables, the constant observational noise variance level for Gaussian likelihood functions of decoders are set to be 0.02 (except for MIMIC dataset where we have used 0.3). During evaluation, we use importance sampling with 10K samples to estimate the log-likelihoods for conditional data generation.

## C.3. Additional experiment pipeline setup

During training of all models, the range of all variables is scaled to be between 0 and 1. This transformation is removed when making predictions on the target variables.

In Section 3.3, to train these partial models on data with missing values, we randomly sample 90% of the dataset to be the training set, and assume that a random fraction (uniformly sampled between 0% and 99%) of feature values are missing on each epoch during training. Then, during test time, we assume that 50% of the test set is observed, and use generative models to infer the unobserved data.

## D. Additional Plots on Bank dataset

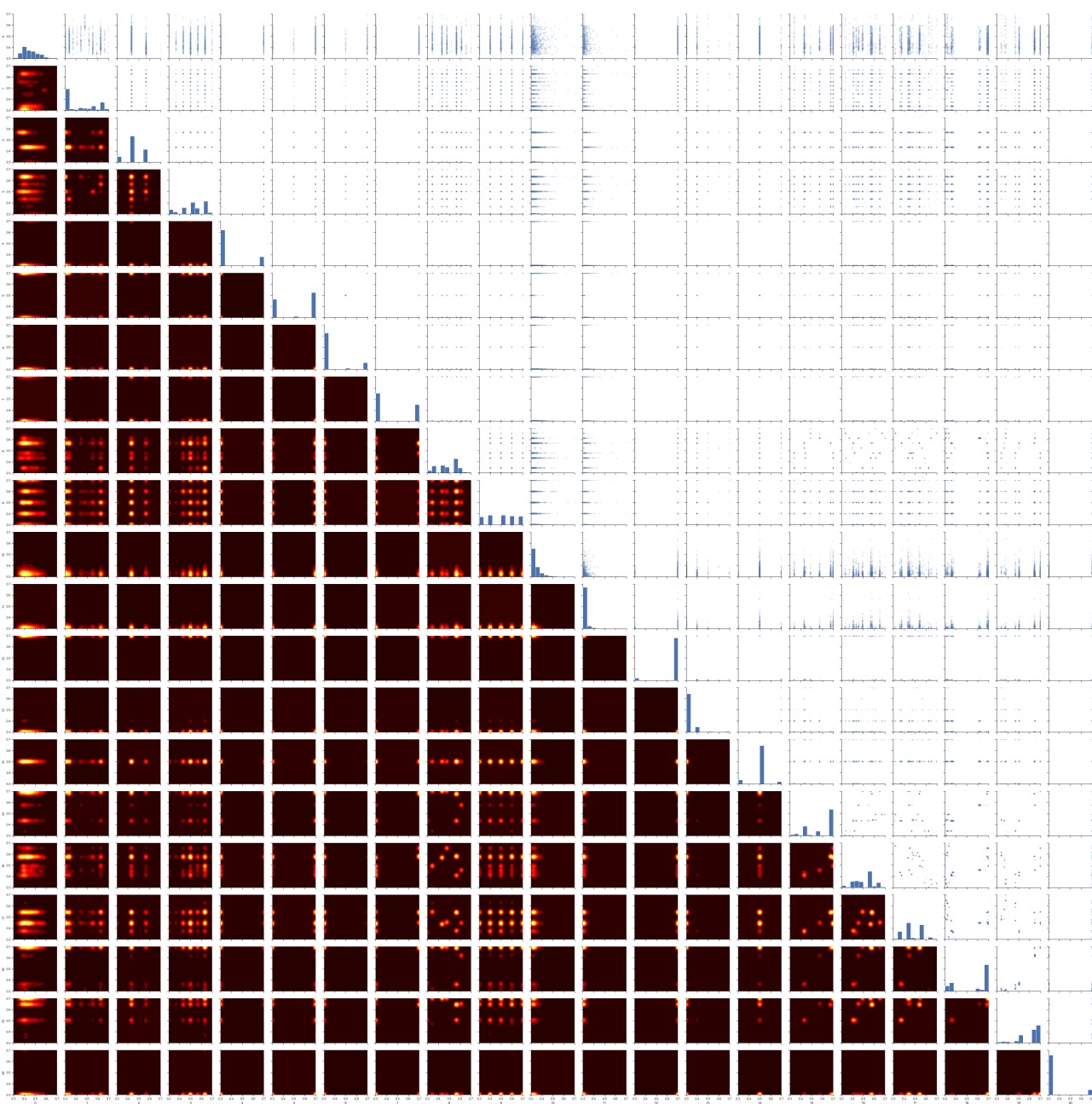

*Figure 3*. pair plots of all variables from the real Bank dataset. Diagonal plots show marginal histograms for each variable. The upper-triangular part shows sample scatter plots for each variable pair. The lower-triangular part shows heat maps identifying regions of high-probability density for each variable pair. For visualization, categorical variables are mapped to a grid of evenly spaced points in the interval $[0, 1]$.

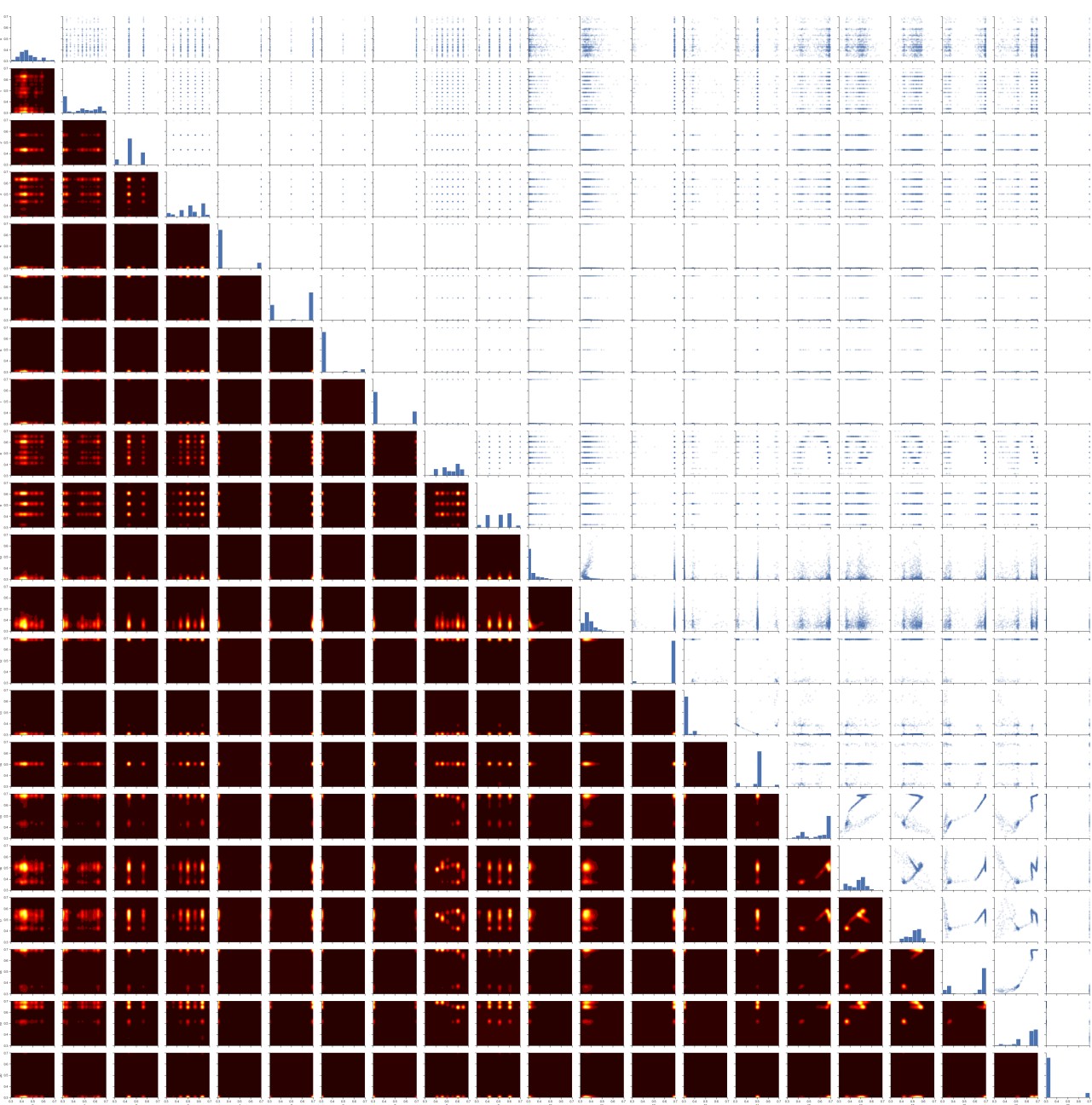

*Figure 4.* pair plots of all variables generated by VAEM. Diagonal plots show marginal histograms for each variable. The upper-triangular part shows sample scatter plots for each variable pair. The lower-triangular part shows heat maps identifying regions of high-probability density for each variable pair. For visualization, categorical variables are mapped to a grid of evenly spaced points in the interval $[0, 1]$.

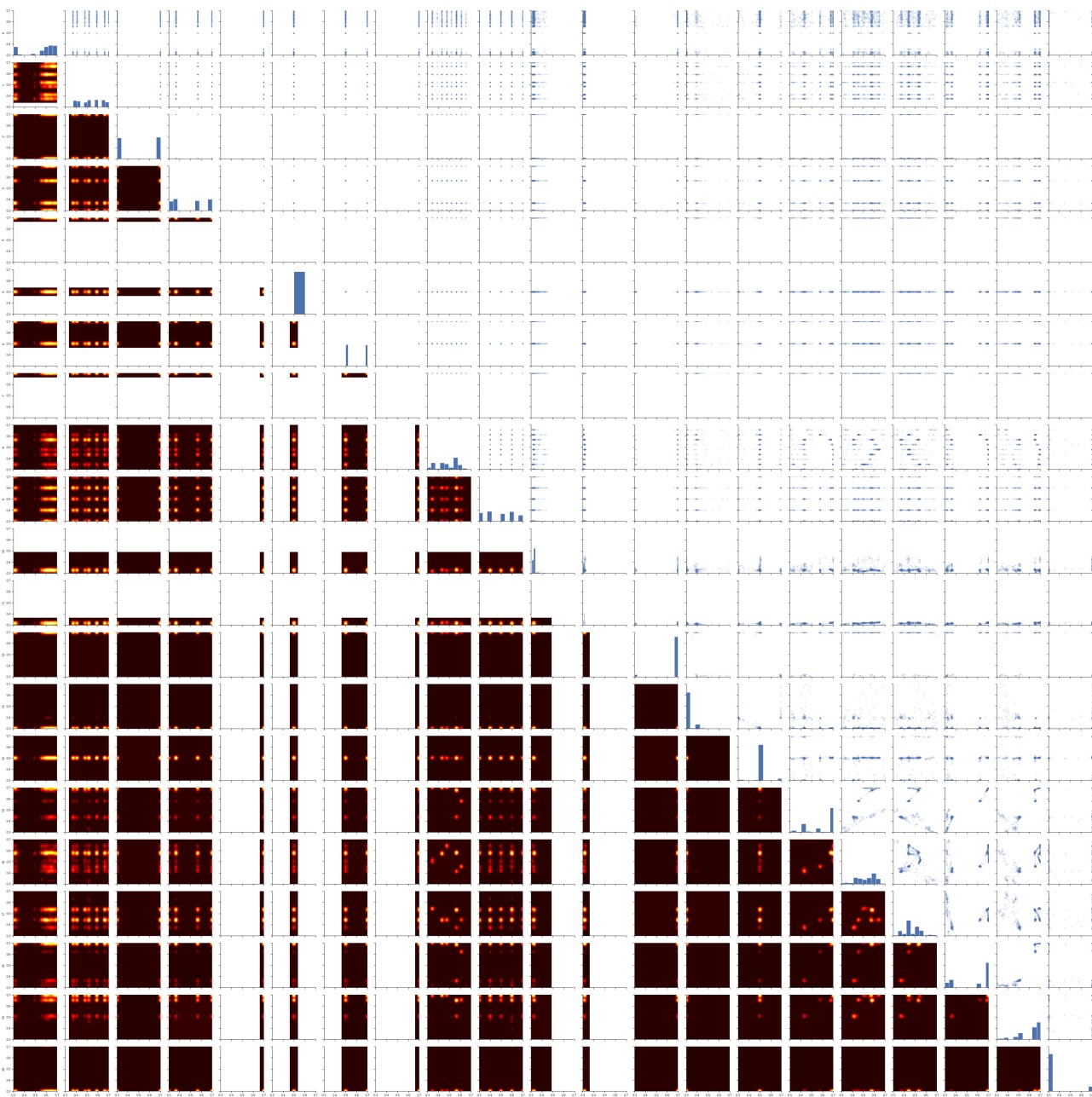

*Figure 5.* pair plots of all variables generated by VAE-balanced. Diagonal plots show marginal histograms for each variable. The upper-triangular part shows sample scatter plots for each variable pair. The lower-triangular part shows heat maps identifying regions of high-probability density for each variable pair. For visualization, categorical variables are mapped to a grid of evenly spaced points in the interval $[0, 1]$.

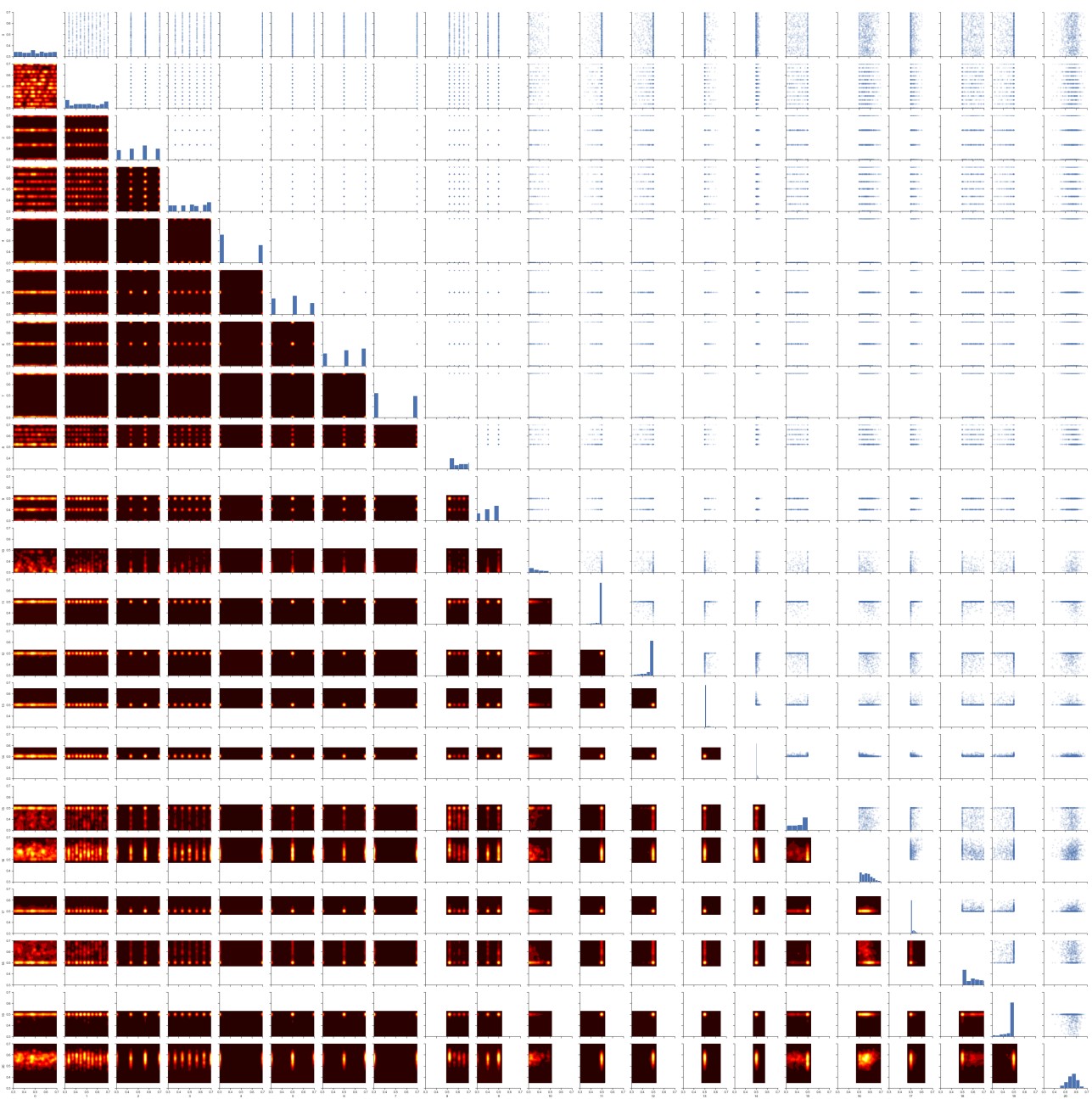

*Figure 6.* pair plots of all variables generated by HI-VAE. Diagonal plots show marginal histograms for each variable. The upper-triangular part shows sample scatter plots for each variable pair. The lower-triangular part shows heat maps identifying regions of high-probability density for each variable pair. For visualization, categorical variables are mapped to a grid of evenly spaced points in the interval $[0, 1]$.

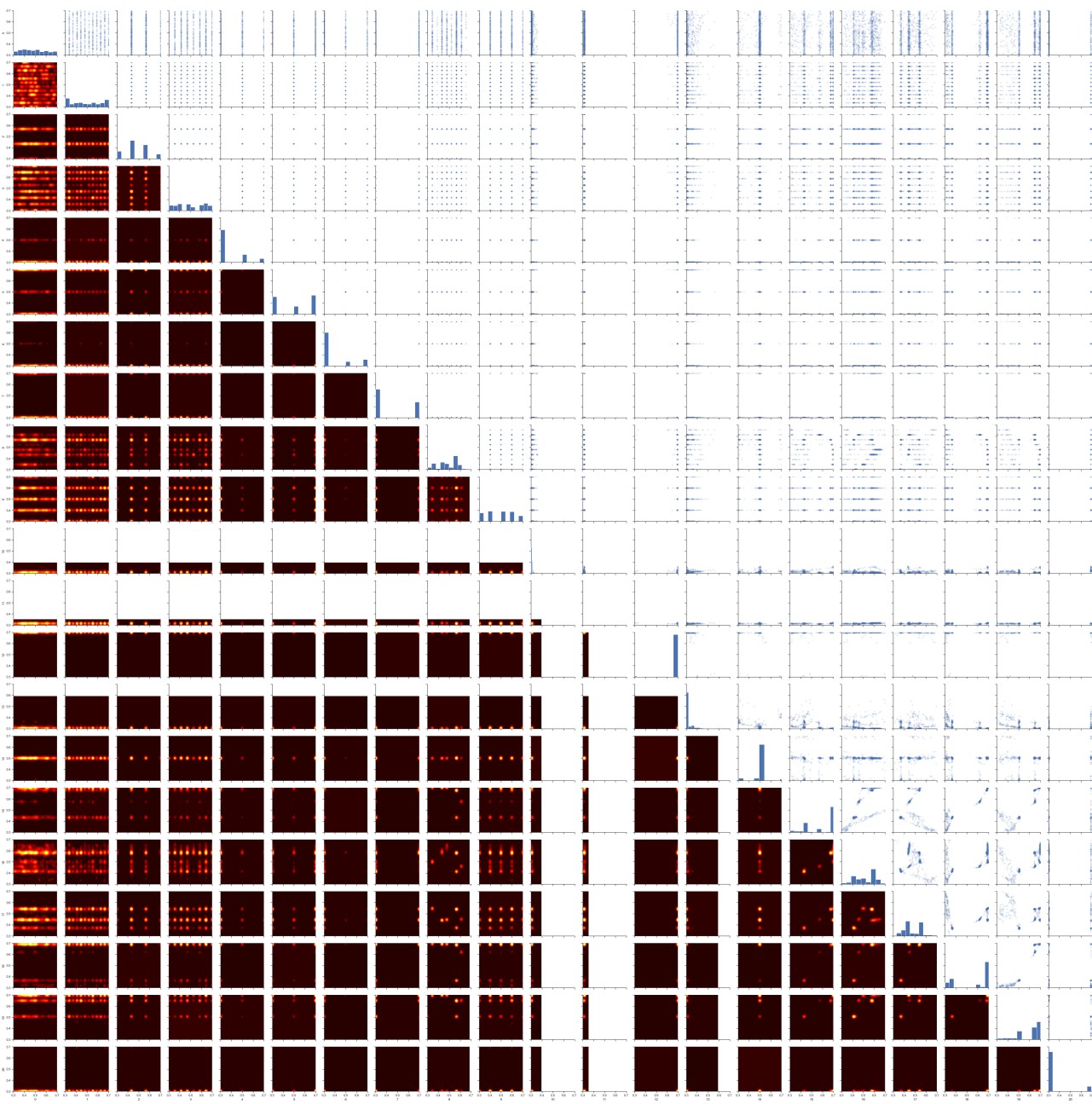

*Figure 7.* pair plots of all variables generated by VAE-extended. Diagonal plots show marginal histograms for each variable. The upper-triangular part shows sample scatter plots for each variable pair. The lower-triangular part shows heat maps identifying regions of high-probability density for each variable pair. For visualization, categorical variables are mapped to a grid of evenly spaced points in the interval $[0, 1]$.