# OpenReview forum: "VAEM: a Deep Generative Model for Heterogeneous Mixed Type Data"
_ICML.cc/2020/Workshop/Artemiss — ICML Artemiss 2020_

### Official Review · AnonReviewer1 · 2020-06-23
**Review of VAEM: a Deep Generative Model for Heterogeneous Mixed Type Data**

**Confidence:** 4
**Rating:** 9

**Review:**

A method for handling missing data and heterogeneity of features in VAEs is proposed. The proposed model is a two-stage model for mixed-type data with missing. Since heterogeneity demands the use of different likelihood functions where one might dominate the other during optimization, for each feature a distinct VAE is fitted to get a latent representation of the feature values and a new VAE is fitted to the latent representations. In order to handle missing data, the permutation invariant approach to the inference network is used in the high-level VAE.

A clear paper with a nice idea that seems to work well in practice, handling the very relevant issue of heterogeneity of features and missing data in VAEs.

Comments:
- The imputations are evaluated using the test-set NLL on the missing data. The imputation error itself could also be reported here.
- A comparison to a VAE with two stochastic layers, like the proposed model, could be included for comparison.

---

### Decision · Program_Chairs · 2020-07-02

**Decision:**

Accept

**Comment:**

We are very happy to inform you that your paper has been accepted for the Artemiss workshop. We will contact you soon to inform you about the details concerning the format of your presentation at the workshop, and the camera-ready version deadline. Please take into account the referee's comments to write the camera-ready version.